# DMPK is a New Candidate Mediator of Tumor Suppressor p53-Dependent Cell Death

**DOI:** 10.3390/molecules24173175

**Published:** 2019-09-01

**Authors:** Katsuhiko Itoh, Takahiro Ebata, Hiroaki Hirata, Takeru Torii, Wataru Sugimoto, Keigo Onodera, Wataru Nakajima, Ikuno Uehara, Daisuke Okuzaki, Shota Yamauchi, Yemima Budirahardja, Takahito Nishikata, Nobuyuki Tanaka, Keiko Kawauchi

**Affiliations:** 1Frontiers of Innovative Research in Science and Technology (FIRST), Konan University, Kobe 650-0047, Japan; 2Mechanobiology Laboratory, Nagoya University Graduate School of Medicine, Nagoya 466-8550, Japan; 3Department of Molecular Oncology, Institute for Advanced Medical Sciences, Nippon Medical School, Tokyo 113-8602, Japan; 4Genome information research center, Research institute for microbial diseases, Osaka University, Osaka 565-0871, Japan; 5Graduate School of Pharmaceutical Sciences, The University of Tokyo, Tokyo 113-0033, Japan

**Keywords:** DMPK, p53, p73, actin, myosin

## Abstract

Tumor suppressor p53 plays an integral role in DNA-damage induced apoptosis, a biological process that protects against tumor progression. Cell shape dramatically changes when cells undergo apoptosis, which is associated with actomyosin contraction; however, it remains entirely elusive how p53 regulates actomyosin contraction in response to DNA-damaging agents. To identify a novel p53 regulating gene encoding the modulator of myosin, we conducted DNA microarray analysis. We found that, in response to DNA-damaging agent doxorubicin, expression of myotonic dystrophy protein kinase (DMPK), which is known to upregulate actomyosin contraction, was increased in a p53-dependent manner. The promoter region of DMPK gene contained potential p53-binding sequences and its promoter activity was increased by overexpression of the p53 family protein p73, but, unexpectedly, not of p53. Furthermore, we found that doxorubicin treatment induced p73 expression, which was significantly attenuated by downregulation of p53. These data suggest that p53 induces expression of DMPK through upregulating p73 expression. Overexpression of DMPK promotes contraction of the actomyosin cortex, which leads to formation of membrane blebs, loss of cell adhesion, and concomitant caspase activation. Taken together, our results suggest the existence of p53-p73-DMPK axis which mediates DNA-damage induced actomyosin contraction at the cortex and concomitant cell death.

## 1. Introduction

Apoptosis is characterized by distinct morphological features, including membrane blebbing, formation of apoptotic bodies, and nuclear fragmentation, all of which require actomyosin contraction [1,2]. In the process of apoptosis, the cysteine-aspartic acid proteases (caspases), critical mediators of apoptosis, are activated, promoting actomyosin contraction. Aberrant activation of myosin II drives not only blebbing but also detachment of cells from the extracellular matrix (ECM), which leads to caspase activation and concomitant apoptosis specially called anoikis [3,4].

Tumor suppressor p53 plays an integral role in apoptosis induced by DNA damage [5,6,7]. In cells under normal conditions, expression of p53 is maintained at a low level via the proteasome system mediated by the E3 ligase Mdm2. Upon DNA damage, p53 is stabilized and activated, inducing transcription of its target genes. Among the transcriptional targets, proapoptotic BH3 proteins such as Bax, Puma, and Noxa powerfully induce apoptosis through mitochondrial outer membrane permeabilization (MOMP) [8,9,10]. MOMP causes the release of cytochrome c and concomitant caspase activation, leading to apoptosis. p73, a p53 family member, induces expression of many genes that are common targets of p53 [11,12]. Like p53, expression of p73 under normal conditions is maintained at a low level via the proteasome system mediated by the E3 ligase Itch [13]. Treatment with DNA damaging agents induces phosphorylation and stabilization of p73, which promotes its transcriptional activity [11,12]. While alternative splicing generates multiple isoforms of p53 and p73 [14,15,16,17], it is well recognized that p53 and p73 proteins induce apoptosis by promoting MOMP. However, it remains unclear whether they induce anoikis by promoting actomyosin contraction. 

DMPK is a serine threonine kinase homologous to myotonic dystrophy kinase-related Cdc42-binding kinase (MRCK) and Rho-Associated Kinase (ROCK) [18]. The myosin-binding subunit of myosin phosphatase target subunit 1 (MYPT1) has been identified as a substrate of DMPK [19,20]. While myosin phosphatase dephosphorylates the myosin II light chain (MLC) and thereby downregulates myosin II activity, phosphorylation of MYPT1 causes inactivation of myosin phosphatase. Therefore, DMPK is expected to induce mysoin II activation. Indeed, overexpression of DMPK increases phosphorylation of MLC 2 in myoblasts [20].

In this study, we show, for the first time, that DMPK expression is induced in response to DNA damage. The DMPK gene is transcribed by p73, which is controlled by p53. An increase in DMPK expression promotes actomyosin contraction, leading to loss of adhesion and concomitant cell death. Our results reveal a novel cell death pathway regulated by the tumor suppressors p53 and p73.

## 2. Results

### 2.1. DMPK is Identified as a Novel p53 Downstream Target Gene Upon Doxorubicin Treatment

It is well recognized that p53 induces apoptosis via MOMP. However, it remains unclear whether p53 induces anoikis by promoting actomyosin contraction. To address this issue, we first conducted a microarray analysis using cDNA from *p53*^+/+^ (wild-type: WT) and *p53*^−/−^ mouse embryonic fibroblasts (MEFs) in the absence or presence of doxorubicin to find a p53 target gene(s) that encodes the modulator of myosin activity. The result showed that doxorubicin treatment induced expression of DMPK in WT MEFs but not in *p53*^−/−^ MEFs. To confirm this result, we examined the levels of DMPK mRNA by qRT-PCR. Consistently, in WT MEFs, but not in *p53*^−/−^ MEFs, expression of DMPK mRNA was increased upon the doxorubicin treatment (Figure 1a). We further evaluated the levels of DMPK protein by western blotting. In WT MEFs, DMPK expression was elevated upon treatment with doxorubicin in a time-dependent manner, while its expression level in *p53*^−/−^ MEFs remained unchanged (Figure 1b). These results suggest that DMPK gene is a transcriptional target of p53.

There are several DMPK isoforms that locate not only in cytoplasm but also in endoplasmic reticulum or mitochondria [20]. Cytoplasmic DMPK isoforms, E and F, are smaller than the other isoforms. Since we detected at least two bands of DMPK in Western blotting, several isoforms of DMPK are likely to be expressed in p53 wild-type expressing cells. To identify which isoform of DMPK was increased by doxorubicin treatment, we performed RT-PCR analysis using primer sets for each isoform. The result showed that the expression of mitochondria- and ER-associated isoforms as well as cytoplasmic isoforms of DMPK was increased by doxorubicin treatment in WT MEFs (Appendix A).

### 2.2. p73, but not p53, Induces DMPK Promoter Activation

To determine whether p53 induces expression of the *DMPK* gene through direct activation of its promoter, we characterized the mouse *DMPK* gene. The transcription initiation site is located at 147 base pairs upstream of the ATG start codon and two potential p53-binding sites, located at –744 to –723 (BS1) and –684 to –663 (BS2), were identified in the *DMPK* promoter region (Figure 2a). The contribution of p53 to *DMPK* promoter activation was examined using a luciferase reporter gene linked to the *DMPK* promoter. To this end, three different plasmids were constructed: DMPK-LP-WT-luc containing p53 potential BS1 and 2, DMPK-SP-WT-luc containing p53-potential BS2 and DMPK-ΔBS-luc, which does not contain any of the two potential p53 binding sites (Figure 2a). Doxorubicin treatment induced the activation of all three reporters (Figure 2b). Unexpectedly, ectopic expression of p53 failed to activate DMPK-LP-WT-luc (Figure 2c), suggesting that p53 activates *DMPK* expression in an indirect manner. It is known that p53 and p73, another member of p53 transcription factor family, share common binding sites in promoter regions [11]. We then investigated whether p73 directly activates *DMPK* promoter. Upon p73 expression, the activity of both DMPK-LP-WT-luc and DMPK-SP-WT-luc, but not of DMPK-ΔBS-luc, was increased, while the activity of DMPK-SP-WT-luc was smaller than that of DMPK-LP-WT-luc (Figure 2c–d). To further examine the role of p73-potential binding sites for *DMPK* promoter activation, we created two constructs, containing a mutation in the BS1 (DMPK-LP-MT-Luc) or the BS2 (DMPK-SP-MT-Luc) (Figure 2a). Ectopic expression of p73 increased the activity of mutated *DMPK* promoters only modestly (Figure 2d). Collectively, these results suggest that p73 directly activates the *DMPK* promoter through binding to the two potential p73-binding sites. Since doxorubicin treatment increased the activity of the DMPK-ΔBS promoter (Figure 2b), transcription factors other than p73 would also be involved in activating the DMPK promoter in response to DNA damage.

### 2.3. p53 is Required for Doxorubicin-Induced p73 Expression

Even though p53 is indispensable for doxorubicin-induced expression of DMPK, the *DMPK* promoter is activated not by p53 but by p73. It is, therefore, possible that p53 regulates DMPK expression indirectly via p73. We next set out to investigate the relationship between p53 and p73. We compared the expression levels of p73 in WT and *p53*^−/−^ MEFs. Expression of p73 was increased upon treatment with doxorubicin when p53 was present (Figure 3a). We further examined whether the p53-dependent p73 expression was observed also in human cells. Consistent with the results in MEFs, downregulation of p53 in MCF-7 cells using shRNA decreased expression of p73 and abolished doxorubicin-induced *DMPK* expression (Figure 3b and Appendix A).

### 2.4. p73 is Required for Doxorubicin-Induced DMPK Expression

Our results shown above suggest that DNA damage activates p53, which upregulates p73 expression, leading to induction of DMPK expression. This regulatory hierarchy was further substantiated using CRISPR based-*p73* gene knockout MCF-7 cells (Figure 4a); the doxorubicin-dependent increase in expression of *DMPK* mRNA was abrogated in *p73*-knockout cells, whereas p73-knockout without doxorubicin treatment did not affect the DMPK expression (Figure 4b). These results indicate that p73 functions downstream of p53 to regulate *DMPK* expression in doxorubicin-treated cells.

### 2.5. Overexpression of DMPK Induces Loss of Cell Adhesion and Caspase Activation

Next, we investigated the consequences of the increase of DMPK expression in MCF-7 cells. Alternative splicing of *DMPK* mRNA generates its multiple protein isoforms, which are differentiated by the presence or absence of the five-amino-acid VSGGG motif and amino acid sequences in the C-terminal region [20]. The C-terminal region of DMPK determines its intracellular localization, while the VSGGG motif modulates its autophosphorylation activity. Among the isoforms, the cytosolic DMPK isoforms, DMPK E and F, reportedly phosphorylate MYPT1 of the myosin phosphatase and thereby inhibit its phosphatase activity, leading to an increase in myosin activity [19,20]. We examined the effect of overexpressing DMPK E, which contains the VSGGG motif, on actin organization and myosin activity in MCF-7 cells. F-actin staining showed that overexpression of wild-type (WT) DMPK caused disruption of cortical actin and formation of blebs at the plasma membrane (Figure 5a–c). In addition, cell rounding and detachment of cells from the extracellular substrate were induced. Phosphorylation of MLC2, a critical step in the activation of non-muscle myosin, was higher at the cortex in DMPK WT overexpressing cells compared with that in non-overexpressing cells (Figure 5a). We then examined whether the kinase activity of DMPK was essential for these effects. Unlike DMPK WT, overexpression of its kinase dead mutant (MT) form induced neither bleb formation nor cell rounding. Myosin activity, evaluated by the phosphorylation level of MLC2, was comparable between DMPK MT expressing cells and non-overexpressing cells (Figure 5a). Since both bleb formation and cell rounding depend on actomyosin contraction [1,2], DMPK would induce these alterations in cell shape though promoting actomyosin contraction. We further examined whether DMPK induced caspase activation using the caspase fluorescent biosensor. When effector caspases (i.e., caspse-3 and -7) are activated, the caspase biosensor translocates into the nucleus [21,22]. In agreement with the effects of DMPK WT-overexpression, but not of DMPK MT-overexpression, on cell shape, the caspase biosensor was accumulated in the nucleus in a much higher extent in DMPK WT-overexpressing cells than in DMPK MT-overexpressing cells (Figure 5d,e). We then examined whether DMPK induced rounding of cells under inhibition of caspases. Treatment of DMPK WT overexpressing cells with the pan caspase inhibitor Q-VD-OPh at 30 h after transfection did not affect rounding of cells (Figure 5f), indicating that caspase activity is not required for DMPK-dependent rounding of cells. Since detachment of cells from ECM induces caspase activation and concomitant anoikis [3,4], our results raise the possibility that an increase in DMPK expression induces loss of adhesion by promoting actomyosin contraction, which leads to anoikis. Consistently, knockdown of DMPK increased viability of doxorubicin-treated WT MEFs (Appendix A).

## 3. Discussion

In this study, we provide evidence that the p53-p73-DMPK axis is a novel cell death-inducible pathway. Upon doxorubicin treatment, p53 induces DMPK expression through activation of p73. An increase in DMPK expression causes cell detachment from ECM along with actomyosin activation, resulting in anoikis. Since ultraviolet (UV) irradiation also increased DMPK expression (Appendix A), p53-mediated DMPK expression might be a common response against DNA damage.

Overexpression of cytoplasmic DMPK induces formation of blebs in both lens epithelial cells [23] and MCF-7 cells (this study). It has also been shown that overexpression of cytoplasmic DMPK induces formation of thick stress fibers in myoblasts [24]. Both blebbing and stress fiber formation depend on myosin activity [1,2]. Thus, increased expression of DMPK would induce remodeling of the actin cytoskeleton by activating myosin, even though the resultant cytoskeletal architecture may be different depending on the cell type.

In this study, we show that depletion of p53 downregulates doxorubicin-induced expression of p73 (Figure 3 and Appendix A). In addition, we observed that p73 expression is decreased by depletion of p53 without doxorubicin treatment (Appendix A). However, it is controversial whether p53 regulates p73 expression under the normal condition, because it has been reported that overexpression of p53 increases p73 expression, on one hand, and p73 expression is upregulated upon depletion of p53, on the other hand [25]. Expression of another p53 family protein p63 was not decreased upon p53 depletion in doxorubicin-treated cells (Appendix A), suggesting that p63 is not involved in the p53-mediated DMPK expression.

Like p53, p73 also has tumor suppressive activity [11,12]. Indeed, it has been revealed that p73 induces cell cycle arrest and apoptosis in cells in which p53 function is abrogated [26]. On the other hand, accumulating evidence shows that loss of p53 function strongly hampers the inhibitory effect of DNA damage on cell proliferation [27,28]. Mutations in p53 gene frequently occur in human cancer, which makes cancer resistant to treatments with DNA damaging agents [29,30]. Pharmaceutical or genetic activation of p73 in cancer cells with loss of p53 function would provide a novel therapeutic strategy for cancer treatment which overcomes chemotherapy resistance.

In addition, our findings will contribute to understanding the mechanism of onset of myotonic dystrophy type 1 (DM1), which is the most common adult form of muscle dystrophy. Expression of CUG triplet repeat expansion in the 3’UTR of *DMPK* mRNA is associated with the disease state of DM1 [31,32]. In DM1 patients, the *DMPK* mRNAs with expanded CUG triplet repeats in the 3´UTR form foci in the nucleus and act as toxic RNA molecules, causing misregulation of splicing in multiple genes [33]. While DM1 is defined as a disease that causes progressive muscular weakness and atrophy, DM1 patients have an increased cancer risk in many organs and exhibit whole-body insulin resistance [34,35]. Even though its expression is relatively high in muscle tissues, i.e., skeletal, cardiac, and smooth muscles, *DMPK* is expressed in various tissues [36,37,38,39]. DNA damage would promote the formation of the toxic RNA foci via induction of *DMPK* expression in various tissues, which might underlie the fact that these symptoms are observed in many body systems. Pharmaceutical inhibition of *DMPK* expression and/or RNA foci formation may provide a way to delay onset and progression of DM1 disease. It would also be important for DM1 patients to avoid being exposed to chemical and mechanical stimuli that cause serious DNA damage. 

## 4. Materials and Methods 

### 4.1. Cell Culture and Plasmidss

Cells were cultured in Dulbecco’s modified Eagle’s medium supplemented with 10% fetal bovine serum and 1% penicillin/streptomycin for MCF-7 human breast cancer cells, C2C12 mouse myoblasts, and 293T human embryonic kidney cells or 50 μg/mL kanamycin for MEFs. *p53*^+/+^ MEFs and *p53*^–/–^ MEFs were prepared as described previously [40]. The DNA encoding HA-p73 was obtained by PCR from pEGFP p73 expression vector, a gift from Dr. Toshinori Ozaki, and subcloned into the pcDNA3 vector (Invitrogen. Inc., Carlsbad, CA). The DNA encoding wild-type HA-DMPK was obtained by PCR using the MEFs cDNA pool and subcloned into the pcDNA3 vector. The K100A mutation, where Lys 100 of DMPK was replaced by alanine, was then introduced to produce HA-DMPK kinase dead. The plasmid encoding Caspase-3 sensor was a kind gift from Dr. Hu Zhaoyong. To generate retroviruses encoding short hairpin RNA (shRNA) against human *p53* or mouse *DMPK*, the *p53* target sequence, 5´-GACTCCAGTGGTAATCTAC-3´, or the *DMPK* target sequence, 5´-GCCAAGTGTATGCCATGAA-3´, was cloned into a pSuper retro puro (Oligoengine, Seattle, WA, USA). To generate lentiviruses encoding guide RNAs (gRNAs) against *GFP* (control) and human *p73*, the *GFP* target sequence 5´-GGAGCGCACCATCTTCTTCA-3´ and the *p73* target sequence 5´-GCCCTATGAGCCACCACAGG-3´ were cloned into the LentiCRISPR v2 puro (Addgene, Cambridge, MA, USA). Retroviral infection and lentiviral infection were performed as described previously [41,42]. Infected cells were selected using 1.5 μg/mL puromycin for three days.

### 4.2. Antibodies and Materials

Anti-DMPK rabbit polyclonal (Life Technologies, Carlsbad, CA, USA), anti-α-tubulin mouse monoclonal (DM1A; Sigma-Aldrich, St. Louis, MO, USA), anti-p53 mouse monoclonal (1C12; Cell Signaling Technology, Danvers, MA, USA and DO1; Santa Cruz Biotechnology, Santa Cruz, CA, USA), anti-p73 mouse monoclonal (ER-15; abcam, Cambridge, MA, USA), and anti-p73 rabbit monoclonal (D3G10; Cell Signaling Technology, Danvers, MA, USA) antibodies were used for immunoblot analysis. Anti-HA mouse monoclonal (16B12; Covance, Princeton, NJ, USA), anti-phosphorylated Ser 19 myosin light chain 2 rabbit polyclonal (pMLC2) (Cell Signaling Technology, Danvers, MA, USA) antibodies were used for immunofluorescence analyses. Doxorubicin and Q-VD-OPh were purchased from Calbiochem (La Jolla, CA, USA) and Sigma-Aldrich (St. Louis, MO, USA), respectively.

### 4.3. DNA Micro Array and Quantitative Real-Time PCR (qRT-PCR)

Total RNA was isolated and purified using NucleoSpin RNA kit (Takara Bio Inc., Shiga, Japan). For microarray, total RNA (200 ng) was reverse-transcribed into double-stranded cDNA using AffinityScript multiple temperature reverse transcriptase (Agilent Technologies Inc., Palo Alto, CA, USA). The resulting complimentary RNA (cRNA) were labeled with cyanine-3 (Cy-3)-labeled cytosine triphosphate (Perkin-Elmer, Wellesley, MA, USA) using a Low Input Quick-Amp Labeling kit (Agilent Technologies Inc., Palo Alto, CA, USA). One color experiments were performed by hybridizing four cRNAs onto a Whole Mouse Genome Oligo Microarray ver. 2 (G4846A; Agilent Technologies Inc., Palo Alto, CA, USA). The Subio Platform and Subio Basic Plug-in (v1.12; Subio Inc., Aichi, Japan) were then used to calculate the between-sample fold change. Microarray data have been deposited in NCBI-GEO under accession numbers GSE134201.

For qRT-PCR, cDNA was prepared using PrimeScript 1st strand cDNA Synthesis kit (Takara Bio Inc., Shiga, Japan). qRT-PCR analysis was performed with Thunderbird SYBR qPCR Mix (Toyobo, Osaka, Japan) under the following conditions: 1 min at 95 °C, followed by 40 cycles of 95 °C for 15 sec and 55 °C for 1 min using StepOne Plus Real-Time PCR system (Applied Biosystems). The following primers were used: mouse *DMPK* forward 5´-ATAAGTGGGACATGCTGAAGAG-3´; mouse *DMPK* reverse 5´-CTCATCCTGGAAGGCAAAGT-3´; human *DMPK* forward 5´-CTGGGTGTATTCGCCTATGAA-3´; human *DMPK* reverse 5´-CTCCTTGTAGTGGACGATCTTG-3´, mouse *p73* forward 5´-CAGCCTTTGGTTGACTCCTATCG-3´; mouse *p73* reverse 5´-TGGTTGACGGAGGGCAGTTTGT-3´, mouse *p63* forward 5´- GTATCGGACAGCGCAAAGAACG -3´; mouse *p63* reverse 5´- CTGGTAGGTACAGCAGCTCATC-3´, mouse *PPIA* forward 5´-CCTTGGGCCGCGTCTCCTT-3´, mouse *PPIA* reverse 5´-CACCCTGGCACATGAATCCTG-3´; human *UBC* forward 5´-TGACTACAACATCCAGAA-3´; human *UBC* reverse 5´-ATCTTTGCCTTGACATTC-3´. After normalization against mouse *PPIA* or human *UBC* mRNA, the relative expression levels to control are shown.

### 4.4. Immunoblot Analysis

Immunoblot analysis was performed as described previously [41]. Briefly, cells were solubilized with lysis buffer (50 mM Tris pH 7.4, 150 mM NaCl, 1% Triton X-100, 1% SDS, 10 mM EDTA, 1 mM Na3VO4, 10 mM NaF, and protease inhibitor cocktail (Nacalai Tesque, Kyoto, Japan), sonicated and centrifuged at 20,000 × *g* for 15 min. The supernatants were used as total cell extracts and subjected to sodium dodecyl sulfate-polyacrylamide gel electrophoresis (SDS-PAGE).

### 4.5. Luciferase Assay 

The reporter construct DMPK-long promoter (LP)-wild-type (WT)-luc, DMPK-short promoter (SP)-WT-luc, and DMPK-ΔBS-luc were generated by subcloning the PCR-amplified fragments encompassing the promoter region (−744 to +51, −684 to +51, and −650 to +51) of mouse *DMPK* gene into the pGL3-basic vector (Promega, Madison, WI, USA). DMPK-LP-mutant (MT)-luc and DMPK-SP-MT-luc were constructed by inserting, by PCR, synthetic DNAs carrying altered putative p53-recognition sequence (Figure 2a). The control plasmid phRL-TK (Renilla luciferase reporter) was obtained from Toyobo (Osaka, Japan). Luciferase activity was determined using the Dual-Luciferase Reporter Assay System (Promega, Madison, WI, USA).

### 4.6. Fluorescence Microscopy

MCF-7 cells were transfected with the expression plasmid for HA-DMPK-wild-type (WT) or HA-DMPK-kinase dead mutant (MT). The cells were fixed with 4% PFA, permeabilized with 0.1% Triton X-100, and then blocked with 2% BSA in phosphate buffered saline (PBS), the cells were incubated with anti-HA and anti-pMLC2 antibodies. Alexa Fluor 546-conjugated goat anti-rabbit IgG (Molecular Probes, Carlsbad, CA, USA) and Alexa Fluor 647-conjugated goat anti-mouse IgG (Molecular Probes, Carlsbad, CA) were used as secondary antibodies. Alexa Fluor 488 Phalloidin and DAPI (Vector Laboratories, Inc., Burlingame, CA, USA) were used to stain actin filaments and nuclei, respectively. Images were acquired using a confocal microscope (A1R HD25, Nikon) and then analyzed with ImageJ software (NIH).

### 4.7. Evaluation of Caspase-Activation

Cells were transfected with EYFP-tagged Caspase-3 sensor expression vector together with mCherry-control, -DMPK WT, or -DMPK MT expression vector. At 30 h after transfection, the cells were fixed and Z-stack images with an interval of 1.0 μm were obtained using a confocal microscope (LSM700; Zeiss). The average of fluorescence intensities of EYFP in three areas (3 μm in diameter) of cytoplasm and nucleus in each cell were quantified using ImageJ software (NIH). A cell in which the fluorescence intensity in the nucleus is 1.5 times higher than that in the cytoplasm is defined as a caspase-activating cell.

### 4.8. Morphological Evaluation of Cells

Cells were transfected with mcherry-tagged DMPK WT, or DMPK MT expression vector. At 6 h after transfection, the cells were cultured in the presence and absence of Q-VD-OPh for 30 h and then fixed. Z-stack images with an interval of 3.0 μm were obtained using a confocal microscope (LSM700; Zeiss) and analyzed with. ImageJ software (NIH). A cell having thickness above 9.0 μm is defined as a rounding cell.

## Figures and Tables

**Figure 1 molecules-24-03175-f001:**
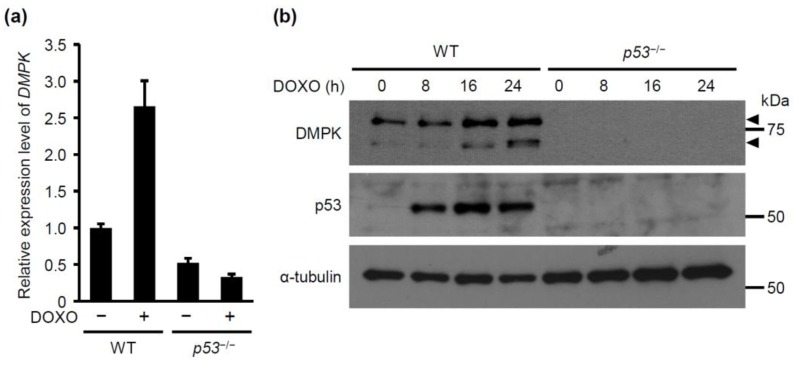
DMPK expression is upregulated upon DOXO treatment of *p53*^+/+^ but not *p53*^−/−^ MEFs. (**a**) DMPK expression in *p53*^+/+^ (WT) and *p53*^−/−^ MEFs treated with or without doxorubicin (DOXO; 0.5 μg/mL) for 24 h was evaluated by qRT-PCR. Each bar represents the mean ± SD; n = 3. (**b**) Lysates from the cells treated with or without DOXO for the indicated time periods were subjected to immunoblot analysis with antibodies against DMPK, p53, and α-tubulin as a loading control. Arrowheads indicate DMPK.

**Figure 2 molecules-24-03175-f002:**
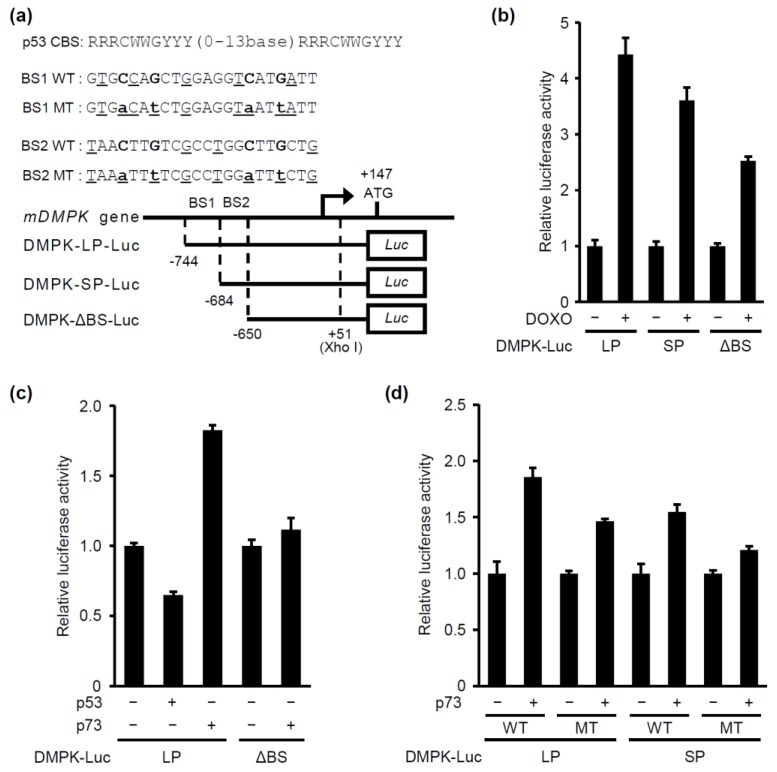
p73 induces *DMPK* promoter activation. (**a**) The location of p53-putative binding sequences within *DMPK* promoter and the luciferase reporter gene constructs. p53-consensus binding sequences (CBS) and p53-putative binding sequences (BS1 and BS2) within *DMPK* promoter are shown. R is A or G, W is A or T, and Y is C or T. The sites difference from the p53 CBS are indicated by underlines. The following reporter plasmids used in this assay are also indicated: DMPK-LP-WT-luc containing the promoter region −744 to +51 of the mouse *DMPK* gene, DMPK-SP-WT-luc containing the promoter region −684 to +51 of the mouse *DMPK* gene, DMPK-LP-MT-luc and DMPK-SP-MT-luc containing the mutations in BS1 or BS2 (indicated by lowercase letters). DMPK-ΔBS-luc containing the promoter region −650 to +51 does not contain the p53-BS1 and -BS2. (**b**) C2C12 cells were transfected using DMPK-LP-WT-luc, DMPK-SP-WT-luc, or DMPK-ΔBS-luc construct and treated with DOXO (0.5 μg/mL) after 12 h. The luciferase activity was measured 24 h after DOXO treatment. The activity was normalized with the average value of no-treated cells. (**c**) Expression plasmid for HA-p53 or HA-p73 was cotransfected with DMPK-LP-WT-luc or DMPK-SP-WT-luc plasmid. (**d**) Expression plasmid for HA-p73 was cotransfected with DMPK-LP-WT-luc, DMPK-LP-MT-luc, DMPK-SP-WT-luc, or DMPK-SP-MT-luc plasmid. For figures (c) and (d), the luciferase activity was measured 24 h after transfection and it was normalized with the average value of the cells transfected with control plasmid for HA-p53 or p73. For figures (b) to (d), each bar represents the mean ± SD; n = 3.

**Figure 3 molecules-24-03175-f003:**
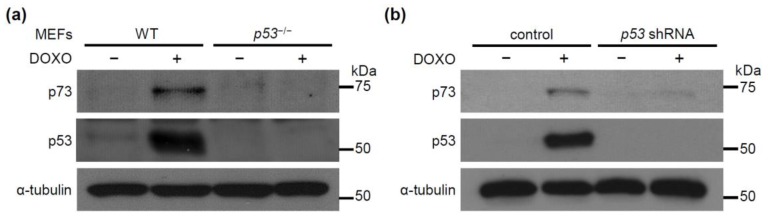
DOXO treatment induces p73 expression in a p53 dependent manner. (**a**) WT (*p53*^+/+^) and *p53*^−/−^ MEFs were treated with DOXO (0.5 μg/mL) for 16 h. (**b**) MCF-7 cells infected either with control or *p53* shRNA-expressing retrovirus were treated with DOXO (1.0 μg/mL) for 16 h. Cell lysates were subjected to immunoblot analysis with antibodies against p73, p53, and α-tubulin as a loading control.

**Figure 4 molecules-24-03175-f004:**
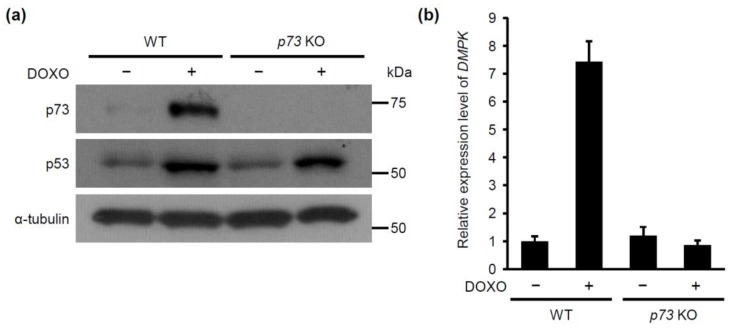
Knockout of *p73* decreases DOXO-induced *DMPK* expression. MCF-7 cells were infected with control or *p73* gRNA-expressing lentivirus using CRISPR/Cas9 system. (**a**) The cells were treated with DOXO (1.0 μg/mL) for 16 h. Subsequently, cell lysates were subjected to immunoblot analysis with antibodies against p73, p53, and α-tubulin as a loading control. (**b**) The cells were treated with DOXO for 24 h. Subsequently, *DMPK* expression level was evaluated using qRT-PCR. Each bar represents the mean ± SD; n = 3.

**Figure 5 molecules-24-03175-f005:**
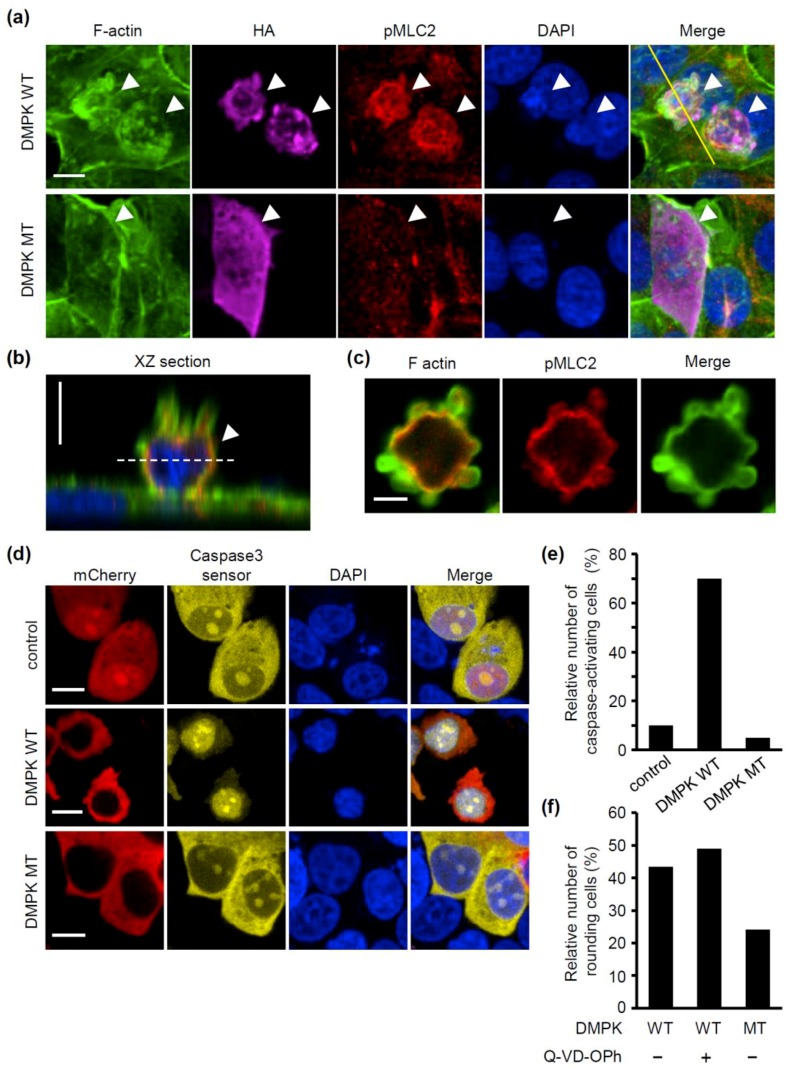
Overexpression of DMPK induces caspase activation in a kinase activity dependent manner. (**a**–**c**) MCF-7 cells were transfected with expression plasmid for HA-DMPK-wild-type (WT) or HA-DMPK-kinase dead mutant (MT). Confocal images of cells stained for F-actin using phalloidin (green), HA (purple), pMLC2 (red) and DNA using DAPI (blue). (**a**) Z-stack images with 25 slices at 1.0 μm intervals were obtained. The projected images are shown. Arrowheads indicate DMPK WT or MT overexpressing cells. Scale bar is 10 μm. (**b**) XZ section image along the yellow line of a cell expressing HA-DMPK-WT in figure a is shown. Image for HA staining is omitted for clarity. Scale bar is 10 μm. (**c**) XY section images along the white dots line of a cell expressing HA-DMPK-WT in figure b are shown. Image for DAPI is omitted for clarity. Scale bar is 5 μm. (**d**, **e**) Cells were co-transfected with EYFP-Caspase-3 sensor and mCherry (control), mCherry-DMPK-WT or mCherry-DMPK-MT. (**d**) Confocal images of MCF-7 cells stained for mCherry (red), Caspase-3 sensor (yellow) and DNA using DAPI (blue). Z-stack images with three slices at 1.0 μm intervals of the middle of cells were obtained. The projected images are shown. Scale bar is 10 μm. (**e**) Relative number of caspase-activating cells are shown (n = 20). (**f**) Cells were transfected with mCherry-DMPK-WT or mCherry-DMPK-MT. After 6 h, cells were treated with Q-VD-OPh (100 μM) for 30 h. Z-stack images with an interval of 3.0 μm were obtained. The relative number of rounding cells are shown (n = 190).

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
