# Peer review of "DMPK is a New Candidate Mediator of Tumor Suppressor p53-Dependent Cell Death"

_molecules, 2019, doi:10.3390/molecules24173175_

Round 1

Reviewer 1 Report

Tumor suppressor p53 plays an integral role in DNA damage-induced apoptosis, a biological process that protects against tumor progression. Cell shape dramatically changes when cells undergo apoptosis, which is associated with actomyosin contraction; however, it remains entirely unknown how p53 regulates actomyosin contraction in response to DNA-damaging agents. To identify a novel p53 regulating gene encoding the modulator of myosin, the authors conducted DNA microarray analysis. They found that, in response to DNA-damaging agent doxorubicin, expression of myotonic dystrophy protein kinase (DMPK), which is known to upregulate actomyosin contraction, was increased in a p53-dependent fashion. The promoter region of mouse DMPK gene contained potential p53-binding sequences and its promoter activity was increased by overexpression of the p53 family protein p73, but, unexpectedly, not of p53. Furthermore, they found that doxorubicin treatment induced p73 expression, which was significantly attenuated by downregulation of p53. These data suggest that p53 induces expression of DMPK through upregulating p73 expression. Overexpression of DMPK promotes contraction of the actomyosin cortex, which leads to formation of membrane blebs, loss of cell adhesion, and concomitant caspase activation. Taken together, their results suggest the existence of p53-p73-DMPK axis which mediates DNA damage-induced actomyosin contraction at the cortex and concomitant apoptosis.

This is a very unique study that should be circulated after amendment.  My critiques are as follows.

1. It looks the authors studied p53, p73, and DMPK response only in doxorubicin-treated cells. Was the same results obtained in cells receiving non-DOXO DNA damage-induced apoptosis, e.g. gamma-irradiation, non-DOXO anti-cancer drugs?

2. Are there any roles of p63 in the regulation of p53-p73-DMPK pathway?

3. The authors did not cite recent reviews on p53 and p73 in the introduction. Complicated alternative splicing of p53 and p73 will affect that data.

These are just examples from Pubmed.

p53 and its isoforms in DNA double-stranded break repair.

Zhang YX, Pan WY, Chen J. J Zhejiang Univ Sci B. 2019 Jun;20(6):457-466. doi: 10.1631/jzus.B1900167.

p73 Alternative Splicing: Exploring a Biological Role for the C-Terminal Isoforms.

Vikhreva P, Melino G, Amelio I. J Mol Biol. 2018 Jun 22;430(13):1829-1838. doi: 10.1016/j.jmb.2018.04.034. Epub 2018 May 4.

Alterations of p63 and p73 in human cancers.

Inoue K, Fry EA. Subcell Biochem. 2014;85:17-40. doi: 10.1007/978-94-017-9211-0_2.

Are interactions with p63 and p73 involved in mutant p53 gain of oncogenic function?

Li Y, Prives C. Oncogene. 2007 Apr 2;26(15):2220-5.

4. It seems the importance of p53-p73-DMPK axis which mediates DNA damage-induced actomyosin contraction at the cortex and concomitant apoptosis, is common in mice and humans. Only MCF7 cells have been tested in Figure 5. Did they get the same results in human cells with wild type p53 (HMEC, MCF10A, H460)? Are there any differences in p53-null and mutant cells in DNA damage response?

Reviewer 2 Report

Rademaker et al. convincingly demonstrated that DMPK, myotonic dystrophy protein kinase, which is known to induce actomyosin contraction, is induced by doxorubicin in a p53-dependent manner. They also showed that DMPK is a direct p73 target gene but not direct p53 target gene, and p53 induces p73 leading to DMPK induction. Furthermore, they showed that DMPK overexpression causes caspase 3 activations, membrane blebs, and actomyosin contraction in its kinase activity-dependent manner, suggesting the role of p53-p73-DMPK pathway in regulating DNA damage-mediated cell death. Novel p53-p73-DMPK pathway induction by DNA damage is quite interesting, and data are clear to support authors’ conclusion. I think this paper will attract interest in the field of p53 and DNA damage and be worth for the publication. I only have minor suggestions for this paper before publication.

In Figure 1, loss of p53 significantly reduces DMPK protein levels without doxorubicin treatment. mRNA levels of DMPK also decrease to half by p53 loss. But p73 loss does not seem to affect the levels of DMPK expression in Figure 4b. Does p53-p73-DMPK pathway is only activated after doxorubicin treatment? Although the focus of this manuscript is after DNA damage, it is better to discuss this pathway without DNA damage in discussion section since DMPK1 reduction after loss of p53 in Figure 1b is quite dramatic.

The data of Figure 5d and 5f will confuse readers. Since it is generally considered that cell rounding occurs after caspase activation during apoptosis, it is puzzled why caspase inhibitors do not affect cell rounding at all during apoptosis mediated by DMPK1 overexpression. Does caspase inhibitor can rescue DMPK-mediated apoptosis (or cell death)? Unless authors like to do more experiments, I think it will be safe to change the word from “apoptosis” in the title, abstract, and results section to “cell death” based on current data since authors did not show any apoptosis data between Figure 1-4 using their cell lines. It is not clear whether p53-p73-DMPK pathway is required for doxorubicin-mediated apoptosis.
